# Major bioactive chemical compounds in Astragali Radix samples from different vendors vary greatly

**Bijay Kafle[1], Jan P. A. Baak[2,3]ʘ, Cato Brede[1,4]ʘ ***

**1** Department of Chemistry, Bioscience and Environmental Engineering, University of Stavanger, Stavanger, Norway, **2** Department of Pathology, Stavanger University Hospital, Stavanger, Norway, **3** Dr Med Jan Baak AS, Tananger, Norway, **4** Department of Medical Biochemistry, Stavanger University Hospital, Stavanger, Norway

ʘ These authors contributed equally to this work.
* cato.brede@uis.no

**Data Availability Statement:** All relevant data are within the manuscript and its S1 File.

**Funding:** Prof Jan P.A. Baak (JB) has a commercial affiliation with the company Dr med Jan Baak AS, Tananger, Norway. The funder provided support in

## Abstract

The worldwide traditional Chinese medicine (TCM) herbs sales figures have increased considerably to 50 billion US$ (2018). Astragali Radix (AR) is amongst the most often sold TCM herbs; sales in the European Union (EU) need European Medicines Agency (EMA) approval. However, comparisons of characteristic bioactive molecules concentrations in AR from different EU vendors are lacking. This study uses liquid chromatography-tandem mass spectrometry (LC-MS/MS) with standard addition to evaluate the influence of different sample and preparation types and ammonia treatment on bioactive molecules concentrations in AR. We also compare AR samples from different EU-vendors. Astragaloside IV (AG-IV), ononin and calycosin 7-O-β-D-glucoside concentrations were higher in root powder samples when extracted with boiled water than with ultrasonication using 70% methanol. AG-IV content was by far the highest in granulates from vendor 1 (202 ± 35 µg/g) but very low in hydrophilic concentrates from vendor 1 (32 ± 7 µg/g) and granulates from vendor 4 (36 ± 3 µg/g). Ammonia-treatment significantly increased AG-IV concentrations in all samples (e.g., to 536 ± 178 µg/g in vendor 1 granulates). Comparable effects were found for most other bioactive molecules. AG-IV and other bioactive molecules concentrations differed strongly depending on sample types, extraction processes, ammonia treatment-or-not and especially between different vendors samples. Ammonia-treatment is debatable, as it is supposed to convert other astragalosides, to AG-IV. The results indicate that routine quantitative analysis of major bioactive compounds present in AR, helps in quality control of AR and to guarantee the quality of commercial products.

## Introduction

The annual worldwide sales figures for traditional Chinese medicine (TCM) herbs have tremendously increased over the past decade to about 50 billion US$ in 2018 [1] and is expected to further increase significantly up to 115 billion in 2025 [2]. TCM herbs are used to maintain

the form of salaries for JB but did not have any additional role in the study design, data collection and analysis, decision to publish, or preparation of the manuscript. The specific roles of JB and co-authors are articulated in the 'author contributions' section.

**Competing interests:** Prof Jan P.A. Baak (JB) has a commercial affiliation with the company Dr med Jan Baak AS, Tananger, Norway. This does not alter our adherence to PLOS ONE policies on sharing data and materials.

and strengthen health, prevent and treat diseases and also to prevent serious side effects caused by western medicines such as chemotherapy [3].

Astragali radix (AR), the dried roots of *Astragalus mongholicus* Bunge and *Astragalus propinquus* Schischkin, family Leguminosae, (known in China as Huang Qi) is one of the strongest and most widely used herbs of the subgroup "tonifying TCM herbs". AR is regarded by many TCM specialists as one of the most important and often used herbs. Indeed, there is a broad therapeutic spectrum for AR, such as improving survival, strongly reducing the side effects of chemotherapy and improving the quality of life (QOL) of metastatic non-small cell lung cancer patient [4, 5] and colorectal cancers of any stage [6, 7]. Other experimental studies claimed that AR possesses many different biological qualities, such as anticancer, anti-virus, anti-asthma, protection from radiation [8], antioxidant capacity, immunomodulatory effects [9], anti-oxidative, cardiovascular and liver protection [10].

The potential bioactivity of AR is due to the presence of bioactive compounds such as isoflavonoids, saponins, polysaccharides, amino acids and trace elements [8] of which astragalosides (AG) (I-VIII) are considered characteristic bioactive therapeutic compounds. Astragaloside saponins showed anti-aging, anti-inflammatory and anti-tumor effects [11], can act as an anti-cancer agent by targeting the non-steroidal anti-inflammatory drugs-activated gene (NAG-1) during its regulation of apoptotic activities [12] and inhibit cell proliferation in human colon cancer cell lines and tumor xenografts [13]. Astragaloside IV (AG-IV) showed potent anti-hepatitis B virus activity [14] and inhibited the growth, invasion and migration of lung cancer cell [15]. Experiments on cycloastragenol (an aglycone of AG-IV), revealed that it can delay the cellular aging process by increasing telomerase activity [16]. Treatment of neurodegenerative diseases can be facilitated by use of cycloastragenol because of its telomerase activating and cell proliferating properties [17]. Cycloastragenol can be produced metabolically by intestinal bacteria or by acid hydrolysis of AG-IV [17, 18] and even by metabolic conversion of other astragalosides (AG-I, AG-II and AG-III) [19]. However, studies of AR as a single herb are nearly exclusively performed in cell cultures and animal models [20–22].

To the best of our knowledge, there have only been a few studies on the pharmacological effects of AR in humans, including renal protection effects [23] and against sudden hearing loss [24]. Formononetin and AG-IV have been tested in healthy human volunteers and appeared totally safe. Pharmacokinetic analysis of formononetin in healthy Chinese volunteers (n = 9), showed the highest plasma concentration of 2.39 ± 1.20 ng/mL 2.15 hours after ingestion of 30 g ultrafine granular powder of AR [25]. Moreover, intravenous infusion of astragaloside IV was well tolerated in a single and in multiple administration(s) of 54 mg: without accumulation of astragaloside IV in plasma. Urine excretion was not the major route of elimination [26], probably metabolically transformed.

Despite lacking scientific evidence for effects in humans, AR has been used in China for hundreds of years. The pharmacopeial recommendations are therefore based on extensive empirical experience. The daily therapeutic dosage of AR advised in humans varies from 9–30 g [27]. Side effects are said to be minimal, if any. It is also allowed in pregnancy and to our knowledge no publications of side effects in pregnancy are known. Yet, these empirical and clinical practices are not in agreement with the aim and statement, that the selection of TCM herbs by TCM practitioners should be in line with scientific standards and management systems.

Assays for quality control of AR are described in Chinese, European & Taiwanese pharmacopoeias. Previously, the amount of AG-IV after assay present in AR, should not be less than 0.04% (w/w) to pass the quality test [28]. However, he recent edition of Chinese pharmacopoeia 2020 stated, that the threshold of AG-IV must not be less than 0.08% (w/w). The pharmacopoeias also advise that the pharmacopeial assays should use ammonia during extraction. However, ammonia is supposed to convert many astragalosides (without proven biological

effect) into AG-IV [29]. With oral intake by humans, such a conversion may not occur. To reach the 0.08% acceptance limit for AG-IV, will therefore also depend on the concentration of other non-AG-IV astragalosides, but does not say anything about it´s true AG-IV biological activity.

In Europe, the 2004/24/EC directive includes herbal medicinal products. TCM products sold in the European Union (EU) [30], need approval by the European Medicines Agency (EMA) and adherence to their guidelines [31]. Despite the abovementioned shortfall in scientific evidence, AR is allowed by the EMA and sold in the EU.

Inevitably, the phytochemical composition of the herbs grown naturally varies due to differences between plant origin, geographical conditions and growth environment [9, 32]. There are also differences between different Astragalus species [33]. Post-harvesting factors such as processing and preparation of the herbs, may further influence the final composition of bioactive components [10, 34]. It therefore could well be, that AR sold by different vendors in the EU for medical purposes, vary in quality and composition. There are many products of AR available in the market of EU in which the composition of bioactive compounds is not stated. AR prescriptions used by different EU health care providers may therefore vary, be suboptimal and lead to suboptimal therapeutic effects. Quantification of bioactive molecules in AR samples is essential. It is also of great importance to know the effects of the factors influencing the composition of the final AR product, to know the real amounts of phytochemicals ingested in such experiments.

In the present study, we investigated whether there is any variation in the concentrations of major isoflavones and AG-IV in different AR herbal types (raw herbs, granulates, tablets, and hydrophilic concentrates) sold in the EU. These commercial samples are widely used in the European Union and are indeed representative for TCM applications. Furthermore, we assessed the concentrations of phytochemicals in dry root samples when using two different extraction methods (conventional boiling in water and a much faster technique sonication using 70% methanol). We also investigated how the measured concentration levels were affected by adding ammonia solution to the extracts. Finally, we analysed the quantity of these bioactive molecules in AR sold by different EU vendors.

Different methods have been used to determine major bioactive isoflavonoids and saponins present in AR samples using ultraviolet (UV) and tandem mass spectrometry (MS/MS) detection after liquid chromatographic separation [35, 36]. Quantification in the present study was performed with a validated LC-MS/MS method as previously described [36]. The method utilized standard addition quantification to overcome the issue of lack of isotopic labelled internal standards. This method greatly improves the measurement accuracy of phytochemicals in AR herbs.

## Materials and methods

### Chemicals and reagents

The reference standard chemical compounds of Formononetin ($\geq$ 98%, lot no: BCBZ9069) and Cycloastragenol ($\geq$ 98%, lot no: SLBM2014V) were purchased from Sigma-Aldrich Co. (St. Louis, USA). Astragaloside IV (98%, lot no: PRF90922502), Ononin (98%, lot no: PRF9060501) and Calycosin 7-O-β-D-glucoside (98%, lot no: PRF8071905) were purchased from Chengdu Biopurify Phytochemicals Limited (Sichuan, China). Molecular structures of these major bioactive compounds are provided (Fig 1). Methanol, acetonitrile and formic acid were of LC-MS grade obtained from VWR International AS (Oslo, Norway). All other reagents were of analytical grades. Purified water was prepared using Elga-purelab Flex water purification system (High Wycombe, UK).

A)

B)

C)

D)

E)

**Fig 1.** Molecular structures of A) formononetin ($C_{16}H_{12}O_4$), B) ononin ($C_{22}H_{22}O_9$), C) calycosine 7-O-β-D-glucoside ($C_{22}H_{22}O_{10}$), D) astragaloside IV ($C_{41}H_{68}O_{14}$) and E) cycloastragenol ($C_{30}H_{50}O_5$).

## Sample extracts

The AR samples were purchased from four different vendors as dried roots, granulates, hydrophilic concentrate, capsules or tablets (Table 1). AR granulates (Kaiser Pharmaceutical Taiwan), Dried roots (Pharmaceutical wholesalers) and Hydrophilic concentrate (Conforma NV, Belgium) were obtained from the Natuurapotheek (NA) (vendor 1), Pijnacker, the Netherlands. Similarly, AR capsules (Swanson Health Products, USA) were obtained from Authentic Produce Limited (AP) (vendor 2), Jersey, UK, AR tablets were obtained from Seven Forest (SF) (vendor 3), Seattle, USA, and AR granulates (Green Nature, Hong Kong, China) were obtained from Chinese Medical Centre (CMC) (vendor 4), Amsterdam, the Netherlands.

The dried roots and tablets were crushed to powder using a mortar and pestle. Dry root powder was acquired from decapsulated capsules. Powdered samples were extracted either by boiling the samples in water for 60 min or by ultrasonication using 70% methanol at 40˚C for 60 min using Branson Ultrasonicator (Danbury, USA). To remove impurities, all sample extracts were centrifuged at 4000 rpm for 10 minutes twice using an Eppendorf Centrifuge 5702 (Hamburg, Germany). The sample extracts were dried using an IKA HB 10 evaporator (Ohio, USA). The dried extracts were reconstituted to final volume of 20 mL with pure methanol. The liquid samples of hydrophilic concentrations were analysed without extraction after centrifugation, using a liquid density of 0.83 g/mL.

## Ammonia treatment of sample extracts

To study the effects of ammonia on bioactive chemical constituents of AR, specially AG-IV, sample extracts were analysed after treatment by addition of an equal volume of 20% ammonia solution. All samples were centrifuged at 4000 rpm for 10 minutes, before injection into LC-MS/MS.

## Standard solutions

The reference standards of all five chemical compounds were weighed and dissolved in methanol to make a solution with final concentration of 1 mg/mL. Ononin and formononetin were dissolved by adding 4–5% acetone in methanol and heating gently to 40˚C. The stock solutions

**Table 1. Sample extracts.**

| Sample ID | Sample/Vendors* | Lot no. | Weight | Extraction conditions | Final volume (mL) |
|---|---|---|---|---|---|
| A | Granulates/NA | GR– 77/3/19 | 5 g | Boiled in water | 20 |
| A1 | Granulates/NA | GR– 77/3/19 | 5 g | Ultrasonication | 20 |
| B | Dried root powder/NA | HB– 01350146 | 5 g | Ultrasonication | 20 |
| B1 | Dried root powder/NA | HB– 01350146 | 2.5 g | Boiled in water | 40 |
| C | Hydrophilic concentrate/NA | HC-17J10/V90291 | 5 g | No extractions | 6.009 |
| | | | ~ 6.009 mL | | |
| D | Dried root capsules/AP | C236378 | 5 g | Ultrasonication | 20 |
| E | Tablets/SF ** | G8855 | 5 g | Ultrasonication | 20 |
| F | Granulates/CMC | 20–211231 | 5 g | Ultrasonication | 20 |

**12% Astragalus Root w/w per tablet.

* NA (Natuurapotheek), AP (Authentic Produce), SF (Seven Forest), CMC (Chinese Medical Centre).

of each standard compound were prepared with a final concentration of 0.1 mg/mL in methanol. The standard dilutions for all compounds were: 0.3125, 0.625, 1.25, 2.5, 5, 10 and 20 μg/mL. For standard addition, diluted samples were spiked to compound concentrations of 0, 0.5, 1 and 2 μg/mL.

### LC-MS/MS analysis

The instrumental analysis was performed by a validated method as previously described [36]. Briefly, the LC-MS/MS instrument was an Acquity UPLC coupled with a Quattro Premier XE triple quadrupole mass spectrometer (Waters Corporation, Massachusetts, USA). Phytochemicals were separated on a reverse phase BEH C18 column (100 mm long x 2.1 mm ID) with 1.7μm particle size and 130 Å pore size (Waters) by using a mobile phase gradient of 0.2% formic acid mixed with methanol. Positive electrospray ionization (ESI+) and MRM were applied for detection. Samples were diluted ten times before analysis, and phytochemicals were accurately quantified by standard addition.

## Results

The influence of the extraction process and different sample types were studied first. The AG-IV concentration in dried roots from vendor 1 when boiled in water was significantly higher (almost double) than in methanolic extracts using ultrasonication (63 ± 6 μg/g vs. 32 ± 7 μg/g). However, the concentration of formononetin was lower in boiled water extract (89 ± 6 μg/g) than in 70% methanol extracts using ultrasonication (133 ± 38 μg/g), although the high standard deviation in the latter results show that the effect varied greatly. The concentration of two other isoflavonoids was higher in boiled water extracts than in extracts prepared by ultrasonication (ononin: 126 ± 3 μg/g vs. 49 ± 6 μg/g; calycosin 7-O-β-D-glucoside 384 ± 24 μg/g vs. 118 ± 23 μg/g). This shows the strong influence of the extraction process and solvents used on the concentration estimates of the bioactive compounds. Boiled water extractions, the classical extraction manner showed superiority over (the faster) 70% methanol extraction using ultrasonication, when AG-IV, ononin and calycosin are the target molecules.

As expected, the granulates sample from vendor 1 had much (>3x) higher concentrations of AG-IV than the raw root samples from the same company. Interestingly, almost equal amounts of AG-IV were measured after granulates samples were extracted using boiled water, as with ultrasonication extraction in 70% methanol (200 ± 70 μg/g vs. 202 ± 35 μg/g).

Ammonia treatment resulted in a manifold increase in the concentration of AG-IV in all samples. In fact, ammonia treatment was necessary to approach the required minimum concentration limit (0.08% w/w = 800 μg/g) for AG-IV specified in the Chinese pharmacopoeia (Table 2), and even then, the threshold level was not reached in any of the samples.

In granulates samples from vendor 1, the concentration of AG-IV increased from 202 ± 35 μg/g to 536 ± 178 μg/g after ammonia solution treatment. Similarly, in dried roots

**Table 2. Pharmacopeial limit of compounds to be present in AR.**

| Pharmacopoeia | Compounds | Required minimum concentration limit (% w/w) | Detectors |
|---|---|---|---|
| Taiwan Herbal Pharmacopoeia (2nd Ed. 2016) | Astragaloside IV | 0.04 | ELSD |
| Chinese Pharmacopoeia (2020) | Astragaloside IV | 0.08 | ELSD |
| Chinese Pharmacopoeia (2010) | Calycosin 7-O-β-D-glucoside | 0.02 | ELSD |
| European Pharmacopoeia (7.0, 2011) | Astragaloside IV | 0.04 | ELSD |
| Japanese Pharmacopoeia (17th Ed. 2016) | Astragaloside IV | Not specified | TLC |

* All the samples tested with ELSD used ammonia for extraction.

from vendor 1, the concentration of AG-IV was much higher after ammonia treatment (32 ± 7 μg/g versus 315 ± 137 μg/g). With ammonia treatment, the concentrations of AG-IV were 369 ± 95 μg/g and 306 ± 71 μg/g in dried root samples from vendor 2 and granulates from vendor 4 respectively. In contrast, isoflavonoids concentration did not show a clear trend but fluctuated after treatment with ammonia (i.e., increased in some but decreased in other samples), Table 3.

We also compared samples from different vendors without ammonia treatment, to avoid measuring artificially high levels of AG-IV with no medical significance, and rather obtain levels that would represent the eventual therapeutic effect of the samples with oral intake in humans.

There were huge and significant differences in the concentrations of bioactive molecules in samples from the different vendors 1, 2, 3 and 4. The most important bioactive compound (AG IV) was highest in granulates samples from vendor 1 (202 ± 35 μg/g). The second highest content was nearly 60% lower (in dried roots in capsules, from vendor 2, 78 ± 11 μg/g). Hydrophilic concentrates from vendor 1 and granulates from vendor 4 had very low concentrations. AG IV concentration was only 9 ± 2 μg/g in tablet samples from vendor 3, which was the lowest from all samples. However, this was not surprising, as the vendor indicated in the specifications that the content of AR was very low.

Calycosin 7-O-β-D-glucoside (384 ± 24 μg/g) and ononin (126 ± 3 μg/g) are highest in boiled root extracts from vendor 1 followed by capsules from vendor 2 when extracted by solvent (70% methanol), while formononetin (133 ± 38 μg/g) is highest in roots from vendor 1 when extracted by solvent (70% methanol). The concentration of isoflavonoids in boiled water extracts of granulates from vendor 1 are: formononetin 22 ± 2 μg/g, ononin 41 ± 2 μg/g and calycosin 7-O-β-D-glucoside 241 ± 53 μg/g. The isoflavonoids contents in hydrophilic concentrate from vendor 1 and granulates from vendor 4 were much less than half the concentrations of isoflavonoids present in raw roots from vendor 1. The concentration of all measured compounds in tablets from vendor 3 were very low, but again, this was as expected, because these tablets contain only 12% of astragalus roots present per g tablet material. The obtained concentration of compounds was calculated per total amount of astragalus root present in the tablets, to measure whether they meet the pharmacopeial threshold standards. Cycloastragenol was not detected in any of the samples tested, in agreement with the fact that it is naturally absent in AR [18].

Surprisingly, the granulate samples from vendors 1 and vendor 4 have hugely different concentration of AG-IV (almost 5.5-fold more in vendor 1 granulate). The isoflavonoids concentrations were similar. There were also variations in chemical composition in dried raw roots from vendor 1 and root powder in capsules obtained from vendor 2, i.e., variation in the same type of herbal samples from different vendors. The differences in the concentrations of the bioactive molecules in the samples from different vendors, are summarized in Fig 2.

## Discussion

Over the past decade, there is a strong increase in interest in the use of TCM herbal medicines worldwide, including the EU. TCM herbs can be bought on the internet and then are often inexpensive, whereas EU-approved herbs can be (much) more costly. It can be difficult for medical prescribers to identify high quality commercially available TCM herbal products, when the characteristic bioactive molecules are not known [33], as is the case with all the samples studied, sold by the EU vendors.

Pharmacopoeias describe procedures and analytical methods for quality control (QC). The Chinese pharmacopoeia (2010) specifies only two species of Astragalus as *A. membranaceous*

**Table 3. Concentrations (µg/g) in different samples of isoflavones and astragaloside IV determined by LC-MS/MS using standard addition.**

| Method | LC-MS/MS, Standard addition | |
|---|---|---|
| Sample | 10 x diluted sample extract | 10 x 2 diluted sample extract treated with 20% ammonia |
| **Sample A (NA Granulates, boiled water extractions)** | | |
| Astragaloside IV | 200 ± 70 | NA |
| Formononetin | 22 ± 2 | NA |
| Ononin | 41 ± 2 | NA |
| Calycosin 7-0-β-D-glucoside | 241 ± 53 | NA |
| **Sample A1 (NA Granulates, 70% methanol extraction using ultrasonication)** | | |
| Astragaloside IV | 202 ± 35 | 536 ± 178 |
| Formononetin | 29 ± 4 | 25 ± 5 |
| Ononin | 35 ± 3 | 13 ± 8 |
| Calycosin 7-0-β-D-glucoside | 118 ± 20 | 121 ± 24 |
| **Sample B (NA Dried root powder, 70% methanol extraction using ultrasonication)** | | |
| Astragaloside IV | 32 ± 7 | 315 ± 137 |
| Formononetin | 133 ± 38 | 78 ± 14 |
| Ononin | 49 ± 6 | 36 ± 8 |
| Calycosin 7-0-β-D-glucoside | 118 ± 23 | 216 ± 80 |
| **Sample B1 (NA Dried root powder, boiled water extractions)** | | |
| Astragaloside IV | 63 ± 6 | NA |
| Formononetin | 89 ± 6 | NA |
| Ononin | 126 ± 3 | NA |
| Calycosin 7-0-β-D-glucoside | 384 ± 24 | NA |
| **Sample C (NA Hydrophilic concentrate, no extractions)** | | |
| Astragaloside IV | 32 ± 7 | 104 ± 28 |
| Formononetin | 29 ± 1 | 21 ± 8 |
| Ononin | 36 ± 3 | 7 ± 0 |
| Calycosin 7-0-β-D-glucoside | 121 ± 23 | 47 ± 11 |
| **Sample D (AP Capsules, 70% methanol extraction using ultrasonication)** | | |
| Astragaloside IV | 78 ± 11 | 369 ± 95 |
| Formononetin | 47 ± 2 | 57 ± 8 |
| Ononin | 109 ± 30 | 112 ± 0 |
| Calycosin 7-0-β-D-glucoside | 336 ± 104 | 13 ± 2 |
| **Sample E (SF Tablets, 70% methanol extraction using ultrasonication)** | | |
| Astragaloside IV | 9 ± 2 | 221 ± 67 |
| Formononetin | 5 ± 0 | 6 ± 0 |
| Ononin | 8 ± 1 | 31 ± 23 |
| Calycosin 7-0-β-D-glucoside | 17 ± 2 | 14 ± 1 |
| **Sample F (CMC Granulates, 70% methanol extraction using ultrasonication)** | | |
| Astragaloside IV | 36 ± 3 | 306 ± 71 |
| Formononetin | 54 ± 1 | 84 ± 2 |
| Ononin | 25 ± 3 | 18 ± 4 |
| Calycosin 7-0-β-D-glucoside | 120 ± 4 | 92 ± 11 |

NA = Not Analysed, Sample E (Tablets from Seven Forest contained 12% of astragalus root per 1 g tablet).

(Fisch.) Bge. or *A. membranaceous* var. *mongholicus* (Bge.) Haiso for therapeutical usage [34]. The quality of these herbs may vary with respect to environmental conditions, geography, age of plant and different species [33, 34]. The recent 2020 edition of the Chinese pharmacopoeia

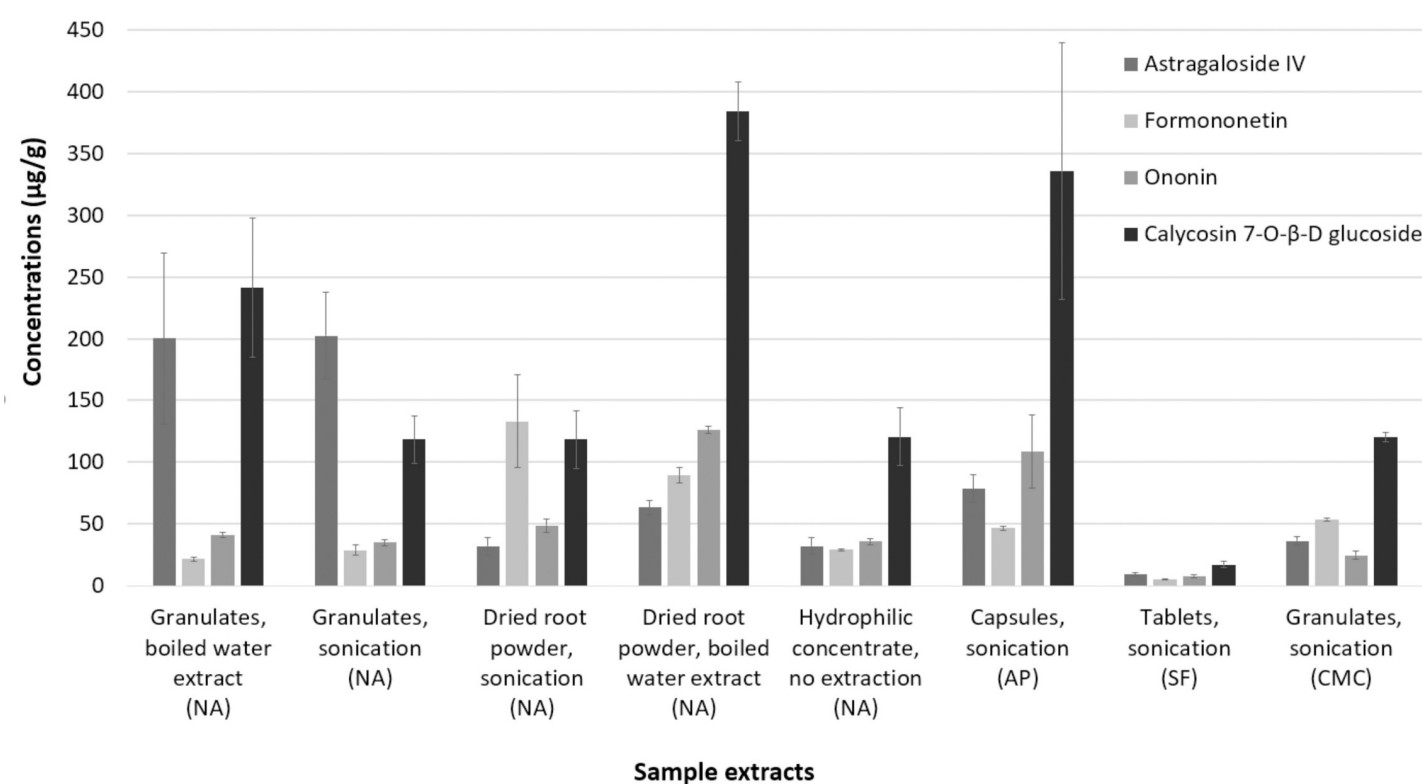

**Fig 2. Histogram showing the concentrations of biologically active components (mean ± SD) present in AR samples from different vendors.**

indicates that AG-IV content should not be less than 0.08% to pass the quality limit. Interestingly, none of the samples studied fulfilled this condition, not even when analysed after using ammonia during sample preparation. These results indicate certain shortcomings of the chromatographic methods prescribed by the pharmacopeias for AR samples [37]. Selection of more reliable detector type is very important for accurate quantification. Studies have used the non-selective ELSD detector which depends upon the retention time of compounds, and since different molecules can co-elute together, it is possible that the identification and quantification of target compounds may be compromised [38]. To avoid these problems, we previously developed a very accurate, sensitive, and reproducible LC/MS-MS method with standard addition quantification [36], which was also used in the present study.

Another debatable point is ammonia treatment of the sample extracts (as recommended in different Pharmacopoeias). Indeed, our results showed that such a process to analyse AR helps to approach the lowest concentration threshold of 0.08% (w/w). One study reported on the analysis of AR samples without ammonia treatment using the standard ELSD detector showing in average of 0.016% (0.08 mg of AG-IV in 0.5 g of AR samples) of AG-IV [39]. These values are similar to the results obtained in our present study without ammonia treatment. The sharp increase in AG-IV concentration by adding ammonia contrasts the much lower concentration of the naturally available characteristic bioactive AG-IV. The increased AG-IV concentration after ammonia treatment therefore is probably due to hydrolytic conversion of other non-AG-IV astragalosides without known medical function [29, 40]. A similar hydrolysis may not occur in the human digestive tract when AR is taken orally. Hence, until we know more about metabolism of different astragalosides in humans, the therapeutic intake levels of AG-IV should be calculated by using ammonia-free extraction measurement results.

Surprisingly, the content of bioactive molecules in AR samples measured by LC/MS-MS using standard addition, varied strongly in samples from different vendors, even when the samples consisted of granulates [9]. In the present work, we applied LC-MS/MS with multiple reaction monitoring (MRM) [22, 41] to ensure a high degree of selectivity and sensitivity in the analysis. Samples were diluted, and the phytochemicals quantified by standard addition, which compensated for ion suppression matrix effects [42, 43] and provided accurate results in the absence of isotopic labelled internal standards [36]. Standard addition therefore provides a viable solution when isotopic labelled internal standards are not available, as is often the case when measuring phytochemicals by LC-MS/MS. There was no cycloastragenol present in any of the commercial samples studied, might be in a lower concentration than the detection limit or naturally absent [18]. Cycloastragenol, which is the aglycon of AG-IV, may be possibly metabolized from AG-IV and other astragalosides, in the human digestive tract [19].

The dried concentrated extracts (granulates) from Vendor 1, are probably easily digested in the human digestive tract, because similar concentrations of AG-IV were found when dissolving granulates in boiling water as with solvent extraction. However, more quantity of bioactive compounds such as AG-IV, ononin and calycosin 7-O-β-D-glucoside were extracted from dried roots using boiled water. This is reasonable because granulates are pre-processed samples. More extensive studies are required to generalize the superiority of traditional boiled water extraction method to other chemical extraction methods. Since methanol and ammonia cannot be used during sample preparation for oral intake, the concentration thus obtained cannot be correlated to the concentration available for systemic absorptions. Granulate samples from vendor 1 have the highest concentration of the most important bioactive molecules. Granulate samples from vendor 4 were cheaper but had 6 times lower bioactive molecules concentrations.

QC studies of TCM herbs will provide baseline data for pharmacokinetics and dose-response relationship during therapeutic use. Simple, preferably non-invasive tests such as analysis of human saliva after administration of TCM samples can be evaluated as a marker for blood concentrations. If these results are positive, saliva testing would make personalized dosaging of TCM herbs in individual patients much easier.

In conclusion, the concentrations of essential bioactive molecules of AR vary greatly between EU vendors. Indeed, the use of ammonia during sample preparation significantly increased the concentrations of AG-IV. Before we know more about hydrolytic conversion of astragalosides in humans, the therapeutic intake levels cannot be estimated from artificially increased testing levels of AG-IV produced by ammonia treatment. Our findings will be relevant for future TCM therapy and for medical research with these herbs.

## Supporting information

**S1 File.**
(XLSX)

## Author Contributions

**Conceptualization:** Bijay Kafle, Jan P. A. Baak, Cato Brede.

**Data curation:** Bijay Kafle.

**Formal analysis:** Bijay Kafle.

**Investigation:** Jan P. A. Baak, Cato Brede.

**Methodology:** Bijay Kafle, Jan P. A. Baak, Cato Brede.

**Project administration:** Jan P. A. Baak, Cato Brede.

**Supervision:** Jan P. A. Baak, Cato Brede.

**Validation:** Bijay Kafle.

**Writing – original draft:** Bijay Kafle, Jan P. A. Baak, Cato Brede.

**Writing – review & editing:** Bijay Kafle, Jan P. A. Baak, Cato Brede.

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
