## [Decision Letter · Decision Letter 0]

17 Mar 2021

PONE-D-20-39234

Major bioactive chemical compounds in Astragali Radix samples from different vendors vary greatly

PLOS ONE

Dear Dr. Brede,

Thank you for submitting your manuscript to PLOS ONE. After careful consideration, we feel that it has merit but does not fully meet PLOS ONE’s publication criteria as it currently stands. Therefore, we invite you to submit a revised version of the manuscript that addresses the points raised during the review process.

MS 'Major bioactive chemical compounds in Astragali Radix samples from different vendors vary greatly' needs minor revision.  Kindly do the needful corrections and submit a revised MS. I also recommend doing English and grammar check before it may be considered for publication in PLOS One. 

We look forward to receiving your revised manuscript.

Kind regards,

Vijai Gupta, PhD in Microbiology

Academic Editor

PLOS ONE

Journal Requirements:

The authors have declared that no competing interests exist.

We note that one or more of the authors are employed by a commercial company: Dr med Jan Baak AS, Tananger, Norway.

 Please also provide an updated Competing Interests Statement declaring this commercial affiliation along with any other relevant declarations relating to employment, consultancy, patents, products in development, or marketed products, etc.  

Additional Editor Comments:

MS 'Major bioactive chemical compounds in Astragali Radix samples from different vendors vary greatly' needs minor revision. Kindly do the needful corrections and submit a revised MS. I also recommend doing English and grammar check before it may be considered for publication in PLOS One.

Reviewers' comments:

Reviewer's Responses to Questions

**Comments to the Author**

1. Is the manuscript technically sound, and do the data support the conclusions?

Reviewer #1: Partly

Reviewer #2: Yes

2. Has the statistical analysis been performed appropriately and rigorously? 

Reviewer #1: Yes

Reviewer #2: Yes

3. Have the authors made all data underlying the findings in their manuscript fully available?

Reviewer #1: Yes

Reviewer #2: Yes

4. Is the manuscript presented in an intelligible fashion and written in standard English?

Reviewer #1: Yes

Reviewer #2: Yes

5. Review Comments to the Author

Reviewer #1: 1. How are the samples selected, are they representative or random?

2. The article discusses that the preparation types may have an impact on the compound, but there is only one method for sample C-F. Does this support the result?

3. The Chinese Pharmacopoeia 2020 edition has been already promulgated and implemented. Astragaloside IV in the current Chinese Pharmacopoeia shall not be less than 0.080%, and the article should be written in accordance with the new standard.

4. Line 280-281, the author mentioned that the vendor indicated in the specifications that the content of Astragali Radix was very low. Does it make sense to choose such a sample?

5. This article mainly talks about the biologically active ingredients of Astragali Radix from different vendors, but there is too much content about the treatment of diseases. It is suggested that the Introduction needs to be revised.

Reviewer #2: This work used LC-MS/MS to evaluate the influence of different sample with standard addition and preparation types and ammonia treatment on bioactive molecules concentrations in Astragali Radix samples from different vendors. I think this is an interesting and meaningful study. Of course, the results of the work are relatively reliable, the discussion is very sufficient and the writing is very formal. I suggest accepting this article.

6. PLOS authors have the option to publish the peer review history of their article (what does this mean?). If published, this will include your full peer review and any attached files.

Reviewer #1: No

Reviewer #2: No

---

## [Author Response · Author response to Decision Letter 0]

7 Apr 2021

We would like to thank both reviewers for their comments and have now revised our manuscript according to the points raised by reviewer #1 as follows (line numbers refer to the clean Manuscript file):

1. How are the samples selected, are they representative or random?

We consciously set out to analyze commercial samples, which are widely used in the European Union. The samples used indeed are representative for this purpose. It would be highly interesting to repeat the study in samples from other continents, such as China.

Sentence added at line 135: “These commercial samples are widely used in the European Union and are indeed representative for TCM applications.”

2. The article discusses that the preparation types may have an impact on the compound, but there is only one method for sample C-F. Does this support the result?

We partially agree on this issue. Samples C-F were prepared by only one type of extraction: Solvent extraction with 70% methanol and sonication. However, for these samples we tested the effect with and without ammonia treatment, so technically we are applying two different preparation methods.

3. The Chinese Pharmacopoeia 2020 edition has been already promulgated and implemented. Astragaloside IV in the current Chinese Pharmacopoeia shall not be less than 0.080%, and the article should be written in accordance with the new standard.

We thank the reviewer for informing us about the new edition. The article has been corrected to include this information, including update of Table 2.

We added new discussion starting on line 307: “Interestingly, none of the samples studied fulfilled this condition, not even when analysed after using ammonia during sample preparation. These results indicate certain shortcomings of the chromatographic methods prescribed by the pharmacopeias for AR samples”

We added reference 37, which specifically mentions chromatographic methods in relation to the 2020 edition of the Chinese pharmacopoeia.

4. Line 280-281, the author mentioned that the vendor indicated in the specifications that the content of Astragali Radix was very low. Does it make sense to choose such a sample?

The purpose of the study was to evaluate the content of Astragali Radix in commercially available samples. It was a conscious choice to select commercial samples, which are widely used in the European Union, independent of the content of Astragali Radix. Our study confirmed that the low content in SF samples is in agreement with the specifications of the manufacturer. This makes the results in our view especially reliable and practically useful. 

5. This article mainly talks about the biologically active ingredients of Astragali Radix from different vendors, but there is too much content about the treatment of diseases. It is suggested that the Introduction needs to be revised.

We beg to differ on this point. In our view, many readers of PLOS ONE are clinically interested and will like such details. However, we have tried to meet the reviewer and have considerably shortened the Introduction.

---

## [Decision Letter · Decision Letter 1]

24 Jun 2021

Major bioactive chemical compounds in Astragali Radix samples from different vendors vary greatly

PONE-D-20-39234R1

Dear Dr. Brede,

We’re pleased to inform you that your manuscript has been judged scientifically suitable for publication and will be formally accepted for publication once it meets all outstanding technical requirements.

Kind regards,

Vijai Gupta, PhD in Microbiology

Academic Editor

PLOS ONE

Additional Editor Comments (optional):

All the editorial, as well as reviewers comments, have been addressed. I recommend the publication of this paper in PLOS One.

Reviewers' comments:

Reviewer's Responses to Questions

**Comments to the Author**

1. If the authors have adequately addressed your comments raised in a previous round of review and you feel that this manuscript is now acceptable for publication, you may indicate that here to bypass the “Comments to the Author” section, enter your conflict of interest statement in the “Confidential to Editor” section, and submit your "Accept" recommendation.

Reviewer #1: All comments have been addressed

2. Is the manuscript technically sound, and do the data support the conclusions?

Reviewer #1: Yes

3. Has the statistical analysis been performed appropriately and rigorously? 

Reviewer #1: Yes

4. Have the authors made all data underlying the findings in their manuscript fully available?

Reviewer #1: Yes

5. Is the manuscript presented in an intelligible fashion and written in standard English?

Reviewer #1: Yes

6. Review Comments to the Author

Reviewer #1: (No Response)

7. PLOS authors have the option to publish the peer review history of their article (what does this mean?). If published, this will include your full peer review and any attached files.

Reviewer #1: No

---

## [Editor Report · Acceptance letter]

28 Jun 2021

PONE-D-20-39234R1 

Major bioactive chemical compounds in Astragali Radix samples from different vendors vary greatly 

Dear Dr. Brede:

I'm pleased to inform you that your manuscript has been deemed suitable for publication in PLOS ONE. Congratulations! Your manuscript is now with our production department. 

Kind regards, 

on behalf of

Dr. Vijai Gupta 

Academic Editor

PLOS ONE